

# Chlorine-36 / beryllium-10 burial dating of alluvial fan sediments associated with the Mission Creek strand of the San Andreas Fault system, California, USA

Greg Balco[1], Kimberly Blisniuk[2], and Alan Hidy[3]

[1]Berkeley Geochronology Center, 2455 Ridge Road, Berkeley CA 94550 USA
[2]Department of Geology, San Jose State University, San Jose, CA USA
[3]Center for Accelerator Mass Spectrometry, Lawrence Livermore National Laboratory, Livermore, CA USA

**Correspondence:** Greg Balco (balcs@bgc.org)

**Abstract.** We apply cosmogenic-nuclide burial dating using the $^{36}$Cl-in-K-feldspar/$^{10}$Be-in-quartz pair in fluvially transported granitoid clasts to determine the age of alluvial sediment displaced by the Mission Creek strand of the San Andreas Fault in southern California. Because the half-lives of $^{36}$Cl and $^{10}$Be are more different than those of the commonly used $^{26}$Al/$^{10}$Be pair, $^{36}$Cl/$^{10}$Be burial dating should be applicable to sediments in the range ca. 0.2-0.5 Ma that are too young to be accurately dated with the $^{26}$Al/$^{10}$Be pair, and should be more precise for middle and late Pleistocene sediments in general. However, using the $^{36}$Cl/$^{10}$Be pair is more complex because the $^{36}$Cl/$^{10}$Be production ratio varies with the chemical composition of each sample. We use $^{36}$Cl/$^{10}$Be measurements in samples of granodiorite exposed at the surface at present to validate calculations of the $^{36}$Cl/$^{10}$Be production ratio in this lithology, and then apply this information to determine the burial age of alluvial clasts of the same lithology. This particular field area presents the additional obstacle to burial dating (which is not specific to the $^{36}$Cl/$^{10}$Be pair, but would apply to any) that most buried alluvial clasts are derived from extremely rapidly eroding parts of the San Bernardino Mountains and have correspondingly extremely low nuclide concentrations, the majority of which most likely derives from nucleogenic (for $^{36}$Cl) and post-burial production. Although this precludes accurate burial dating of many clasts, data from surface and subsurface samples with higher nuclide concentrations, originating from lower-erosion-rate source areas, show that upper Cabezon Formation alluvium is 260 ka. This is consistent with stratigraphic age constraints as well as independent estimates of long-term fault slip rates, and highlights the potential usefulness of the $^{36}$Cl/$^{10}$Be pair for dating upper and middle Pleistocene clastic sediments.

## 1 Introduction: $^{36}$Cl/ $^{10}$Be burial dating

In this paper we apply the method of cosmogenic-nuclide burial dating (henceforth, "burial dating") using the $^{36}$Cl-in-K-feldspar/$^{10}$Be-in-quartz ($^{36}$Cl/$^{10}$Be) nuclide pair to determine the age of buried alluvial sediment exposed in a sedimentary section that has been offset along a strand of the San Andreas Fault in southern California. The concept of cosmogenic-nuclide burial dating relies on a pair of cosmic-ray-produced nuclides that (i) are produced at a fixed ratio by cosmic-ray bombardment of surface rocks and minerals, but (ii) have different decay constants. When sediment is exposed to the surface cosmic-ray flux





during erosion and fluvial transport, concentrations of both nuclides increase over time, but their concentration ratio conforms to the production ratio. If the sediment is then shielded from the surface cosmic-ray flux by burial in a sedimentary deposit, nuclide production halts, and nuclide concentrations decrease due to radioactive decay. If the two nuclides have different decay constants, the ratio of their concentrations will change over time. Accordingly, the ratio of their concentrations can be used

to estimate the duration of burial and hence the depositional age of the sediment. Burial dating has been widely applied to date clastic sediments in caves, river terraces, and sedimentary basins (see summaries in Dunai, 2010; Granger, 2006), and is potentially uniquely useful for Quaternary clastic sediments in arid regions that are difficult to date by other means. In contrast to many other dating techniques that require either fossil material or the formation of new minerals at the time of sediment emplacement, it can be applied to any sediment that has experienced a period of surface exposure followed by burial.

Nearly all existing applications of burial dating have used the $^{26}$Al/$^{10}$Be nuclide pair. These nuclides are produced in quartz at a ratio of $^{26}$Al/$^{10}$Be = 6.75, and the half-lives of $^{26}$Al and $^{10}$Be are 0.7 Ma and 1.4 Ma respectively, so the half-life of the $^{26}$Al/$^{10}$Be ratio is 1.43 Ma. The half-life of the ratio is defined as $-\ln\left(\frac{1}{2}\right)/\lambda_R$, where $\lambda_R = \lambda_A - \lambda_B$ and $\lambda_A$ and $\lambda_B$ are the the decay constants (yr$^{-1}$) of the shorter- and longer-lived nuclides, respectively. The $^{26}$Al/$^{10}$Be nuclide pair has three important advantages for burial dating. First, both nuclides are produced within the same mineral, which ensures that concentrations of

both nuclides reflect the same exposure history. Second, quartz is ubiquitous in clastic sedimentary deposits; third, the mineral chemical composition, and therefore the $^{26}$Al/$^{10}$Be production ratio, is invariant for quartz. However, the relatively long half-life of the $^{26}$Al/$^{10}$Be ratio limits the useful age range of this nuclide pair: it is best suited to burial dating of sediments in the age range 0.7-4 Ma, and is not well suited for middle and late Pleistocene sediments younger than 0.5 Ma (Figure 1). In principle, one can overcome this limitation by using a nuclide pair with a larger difference between half-lives. If the half-life of

the nuclide ratio is shorter, the ratio in a buried sample diverges from the production ratio faster, and a burial age estimate based on the measured ratio is more precise at younger ages. One can show this quantitatively by assuming that a sample experiences a two-stage burial history consisting of (i) a period of surface exposure that is short relative to the half-life of the shorter-lived nuclide, followed by (ii) instantaneous burial at a depth large enough that post-burial nuclide production is negligible. With these assumptions:

$$R_N = \frac{R_M}{R_P} = exp(-\lambda_R t) \qquad (1)$$

where $R_N$ (nondimensional) is the measured nuclide ratio in the sample $R_M$ normalized to the production ratio $R_P$, $\lambda_R$ is the decay constant of the ratio as defined above (yr$^{-1}$), and $t$ is the burial age (yr). Solving for $t$,

$$t = -\frac{1}{\lambda_R} ln(R_N) \qquad (2)$$



If we assume that the half-lives and the production ratio are exactly known, so the uncertainty in a burial age stems only from analytical uncertainty on the measured nuclide ratio, then the relative uncertainty in the burial age $\sigma t/t$ (nondimensional) is:

$$\frac{\sigma t}{t} = \frac{\sigma R_N}{\lambda_R R_N t} \tag{3}$$

where $\sigma R_N$ is the uncertainty in the normalized ratio and is equal to $\sigma R_M/R_P$ where $\sigma R_M$ is the uncertainty in the measured nuclide ratio. Thus, the uncertainty in the burial age is inversely proportional to the decay constant of the ratio $\lambda_R$. Given equivalent precision on the ratio measurements, selecting a nuclide pair with a larger difference between decay constants, and thus a larger $\lambda_R$, results in a more precise burial age. Chlorine-36 has a half-life of 300,000 years, so the difference between the $^{36}$Cl and $^{10}$Be decay constants ($\lambda_R = 1.8 \times 10^{-6}$, corresponding to a half-life of the ratio of 0.38 Ma) is substantially larger

than for $^{26}$Al and $^{10}$Be ($\lambda_R = 4.8 \times 10^{-7}$). Thus, a $^{36}$Cl/$^{10}$Be burial age should be more precise than a $^{26}$Al/$^{10}$Be burial age, as long as the burial duration is not so long that the $^{36}$Cl concentration has decayed to a level that cannot be precisely measured.

    Figure 1 shows the results of a more complex uncertainty analysis (Balco and Shuster, 2009) that also includes a model for measurement uncertainty as a function of nuclide concentration. For the majority of the Pleistocene, $^{36}$Cl/$^{10}$Be burial ages are theoretically expected to be more precise than $^{26}$Al/$^{10}$Be burial ages, and the $^{36}$Cl/$^{10}$Be pair should provide useful (e.g., 20%

or better) precision for burial ages as young as 0.15-0.2 Ma.

    On the other hand, the $^{36}$Cl/$^{10}$Be pair has several disadvantages in comparison to the $^{26}$Al/$^{10}$Be pair. The primary disadvantage is that production of cosmogenic $^{36}$Cl does not occur in pure quartz (SiO$_2$). Although in natural quartz there is sometimes nonzero $^{36}$Cl production from impurities or from Cl in fluid inclusions, the production rate is very low and usually cannot be accurately estimated. Likewise, although $^{10}$Be is produced in other minerals that also have significant $^{36}$Cl production (e.g.,

pyroxene; see Blard et al., 2008; Collins, 2015; Eaves et al., 2018), $^{10}$Be extraction methods and production rates in these minerals are not well established. Thus, measurement of both $^{10}$Be and $^{36}$Cl in the same mineral is generally not feasible. This means that $^{36}$Cl/$^{10}$Be burial dating is only possible when (i) quartz occurs with another mineral that is suitable for $^{36}$Cl measurement, and (ii) both minerals have experienced the same exposure history. Although (ii) might not be the case for individual grains of quartz and some other mineral constituent of sand-sized sediment such as would typically be collected for $^{26}$Al/$^{10}$Be

burial dating, many common geologic situations do meet both requirements, most commonly when alluvial sediment contains granitoid clasts that include both quartz and a potassium or calcium feldspar. Production rates of $^{36}$Cl in these feldspars are comparable to the $^{10}$Be production rate in quartz, and the fact that both minerals are physically part of the same clast ensures that they experienced the same exposure history. In this work we use this approach, of analyzing quartz and potassium feldspars from individual granitoid cobbles.

A second disadvantage is that the chemical composition of feldspars (as well as most other minerals suitable for cosmogenic $^{36}$Cl measurements) is variable, so the $^{36}$Cl production rate is likewise variable. Thus, the $^{36}$Cl/$^{10}$Be production ratio is not invariant across samples like the $^{26}$Al/$^{10}$Be production ratio in quartz, but depends on the chemical composition of the $^{36}$Cl target mineral. In addition, a fraction of total cosmogenic $^{36}$Cl in some feldspar samples can be produced from native $^{35}$Cl in



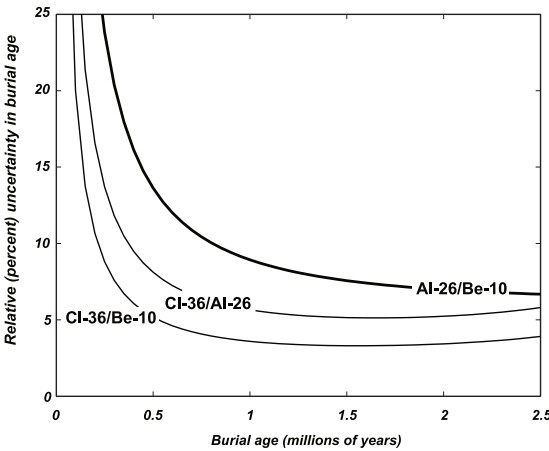

**Figure 1.** Theoretical uncertainty estimate for cosmogenic-nuclide burial ages on Pleistocene clastic sediments computed using the $^{26}$Al/$^{10}$Be, $^{36}$Cl/$^{10}$Be, and $^{36}$Cl/$^{26}$Al pairs. The calculation is described in Balco and Shuster (2009), and includes a model for measurement uncertainty as a function of nuclide concentration fit to typical nuclide concentration measurements. The model assumes that samples experienced a two-stage exposure history consisting of steady erosion at 2 m Myr$^{-1}$ followed by burial at a depth large enough that post-burial production is insignificant. Note that these assumptions are intended to clearly show the contrast between different nuclide pairs, and underestimate the uncertainty expected for some of the samples that we describe in this paper, that originated from areas with much higher erosion rates and thus were buried with much lower nuclide concentrations.

the mineral by capture of secondary thermal neutrons (e.g., Phillips et al., 2001), and the efficiency of this production pathway is affected by the bulk composition and water content of the rock or soil at the site the sample was initially exposed. As it is not possible to measure these properties in sediment source areas for times in the past, this could create a serious obstacle to correctly estimating the production ratio. In this work we use two strategies to mitigate the difficulty of estimating the

5   $^{36}$Cl/$^{10}$Be production ratio during initial exposure of buried clasts. First, we use feldspar separates that have low native Cl concentrations, thus minimizing thermal neutron capture production. Second, we measure $^{36}$Cl/$^{10}$Be ratios in surface samples of granitoids both in outcrop and in modern fluvial sediment upstream of our sample sites. These surface samples have the same lithology as the buried clasts but have not been buried, so their $^{36}$Cl/$^{10}$Be ratios provide a direct estimate of the production ratio in this lithology and can be used to validate production ratio calculations based on rock and mineral chemical composition.



## 2 Geologic context and sample collection

We collected buried granitoid cobbles from two stratigraphic units: the north-dipping Pleistocene Deformed Gravels of White-water (Qd) and the flat-lying Pleistocene Cabezon Formation (Qo), which unconformably overlies Qd (Allen, 1957; Matti et al., 1993; Kendrick et al., 2015). These units are exposed at a ca. 220-meter-thick section in the Mission Creek Preserve, near the

San Gorgonio Pass region of the southern San Bernardino Mountains (Figures 2 and 3). These units form an alluvial fan com-plex that lies to the south of the Mission Creek strand of the San Andreas fault, but is composed of sediment derived from watersheds on the north side of the fault (Fosdick and Blisniuk, 2018). The Qd gravels are distal alluvial fan and ephemeral flu-vial deposits consisting of moderately stratified conglomerate and sandstone. Clast compositions in this unit are predominantly monzonite, granodiorite, biotite gneiss, and hornblende diorite with lesser amounts of phyllite and marble, reflecting basement

metamorphic rocks with intruded Mesozoic plutonic rocks exposed in the San Bernardino and Little San Bernardino Moun-tains. The overlying Qo unit consists of sub-angular to sub-rounded megabreccia fanglomerate and upward-fining, weakly stratified cobble to pebble conglomerate units. Orientations of imbricated cobbles indicate SW and SSE paleoflow directions, which is consistent with the overall geometry of the deposit in showing that the sediment source is to the NE-NNW on the north side of the fault. Clast compositions in this unit are similar to those in the underlying Qd, but the abundance of metasedimentary

and volcanic clasts increases upsection. This coincides with an upsection change in lithofacies, such that very coarse-grained and poorly sorted megablock debris flow deposits of the lower Qo that are rich in biotite gneiss clasts grade upward to weakly stratified debris flow and sheet-flood deposits that have a higher proportion of metasedimentary components.

The fan complex has been transported westward by right-lateral motion along the fault, so successively older sediments are derived from watersheds farther to the east. Although in this paper we focus on determining the age of the alluvial sediments,

the broader objective of the work is to both date the alluvial deposits and match them to source watersheds, thus providing geologic slip rate estimates for this segment of the fault. At present, the fan complex is incised by river canyons crossing the fault at Whitewater Canyon and Mission Creek. Provenance analysis (Fosdick and Blisniuk, 2018) of the buried alluvium comprising the Qo unit that we sampled in this work indicate that upper portions of the Qo alluvium are derived from Mission Creek, lower Qo from Morongo Valley Canyon or Big Morongo Canyon, and upper Qd from unidentified watersheds to the

east of Little Morongo Canyon (Figure 2).

The Qo and Qd units have not been directly dated, but have been mapped as Plio-Pleistocene (Dillon and Ehlig, 1993; Yule and Sieh, 2003; Kendrick et al., 2015) based on comparison with other alluvial fan sediments of the Ocotillo Formation (Keller et al., 1982) located 30 km to the south in the Indio Hills, which are dated to 0.5 - 1.1 Ma (Kirby et al., 2007). Both units must be older than luminescence dates and surface exposure ages of up to ca. 100 ka on inset late Pleistocene alluvial fans of

Mission Creek (Figure 2; Owen et al., 2014; Kendrick et al., 2015). Based on fan provenance and estimated fault slip rates, Fosdick and Blisniuk (2018) estimated that Qo and upper Qd are most likely 200 ka - 1 Ma. Thus, at least the upper portions of Qo are likely to lie in the age range where $^{36}Cl/^{10}Be$ burial dating should be more precise than $^{26}Al/^{10}Be$.

For burial-dating of subsurface alluvial deposits, we collected a total of 11 individual granodiorite clasts, between 12-20 cm in length, from three stratigraphic levels in the Qo alluvium (MCP-6, MCP-7, and MCP-8U) and one level (MCP-8) at



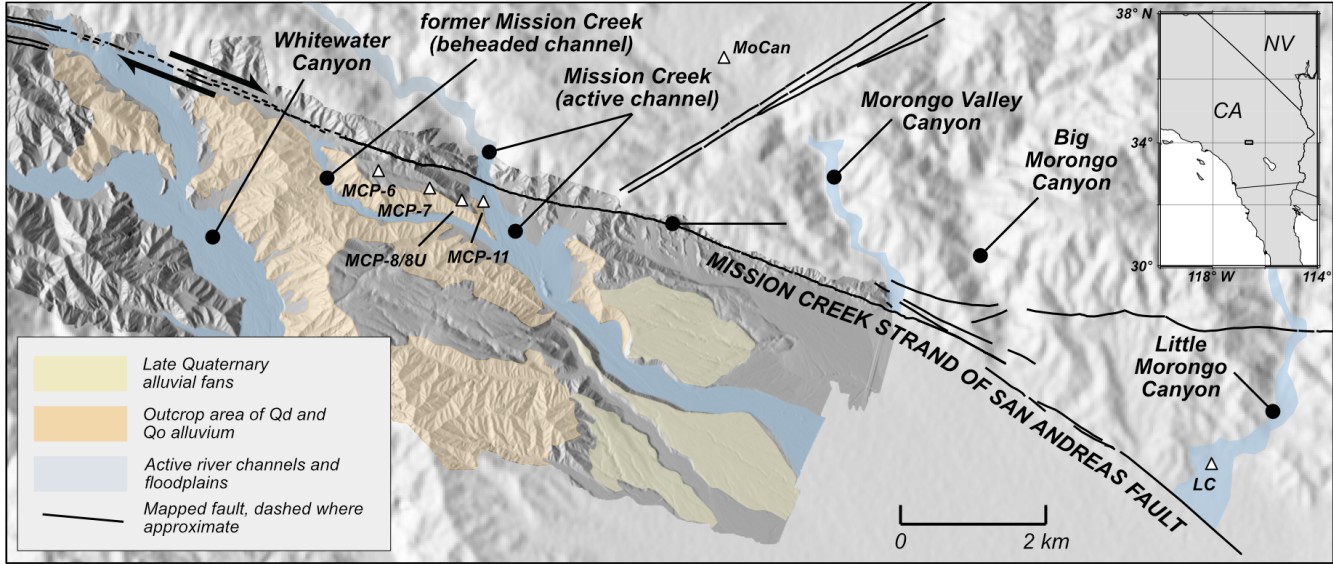

**Figure 2.** Map showing outcrop area of Qo and Qd alluvium and its relation to the Mission Creek strand of the San Andreas Fault. Triangles show sites where surface samples (MCP-11, MoCan, LC) and burial-dating samples (MCP-6, 7, and 8) were collected. Colored regions highlight selected outcrop areas of alluvial deposits and portions of river floodplains discussed in text. The high-resolution shaded-relief topography is from a LIDAR survey by Waco (2017), resampled to 10 m; the lighter background shaded-relief image outside the boundaries of the LIDAR survey is derived from 30-meter SRTM data. Rectangle at center of inset at upper right shows map location.

the top of the underlying Qd deposit (Figures 2 and 4; Table 1). The sample sites follow a channel incising the fan complex that exposes the entire section of Qo alluvium and the uppermost underlying Qd gravels. In principle, collecting multiple clasts from each stratigraphic level should enable us to correct for possible post-burial production using the clast isochron method described by Balco and Rovey (2008) and others (e.g., Granger et al., 2015). However, we also sought to minimize the

5    likelihood of significant cosmogenic-nuclide production during re-exhumation of the samples by excavating 0.25-0.5 m into vertical cliff faces that showed evidence of rapid spalling and retreat (Figure 4). The lithology of the buried clasts we sampled is identical to Cretaceous granodiorites outcropping in source watersheds to the north. Thus, to validate calculations of the $^{36}Cl/^{10}Be$ production ratio in this lithology, we collected three cobbles of the same lithology from modern stream channels crossing the Mission Creek strand ("LC" and "MCP-11" sites in Figure 2), and one sample from the surface of a bedrock tor

10   of this lithology north of the fault (the "MoCan" site).

## 3   Analytical methods

We extracted quartz and potassium feldspar from granitoid clasts in laboratories at San Jose State University by crushing each clast, sieving to 0.25-0.5 mm grain size, and separating a K-feldspar fraction by flotation in a water-based heavy liquid having





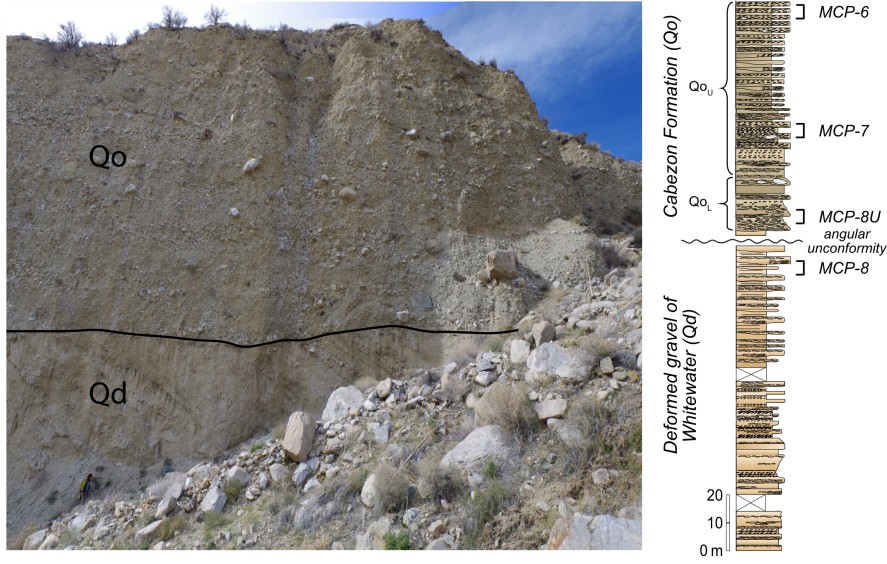

**Figure 3.** Left, annotated photograph of typical outcrop of Qd and Qo alluvium in the field area, showing prominent angular unconformity. Note geologist at lower left for scale. Right, stratigraphic section (Fosdick and Blisniuk, 2018) of Qd and Qo alluvium in the area of sample sites, showing stratigraphic levels sampled (Table 1).

density 2.6 g $cm^{-3}$. We cleaned the K-feldspar fractions in a sonic bath, etched them in hot 10% $HNO_3$ to remove carbonate encrustations, Fe-oxides, and other weathering products, and finally allowed them to soak in 10% $HNO_3$ at room temperature for several weeks. We then rinsed them thoroughly in deionized water before Cl extraction. We extracted and purified quartz from the heavier fraction by a combination of magnetic separation and froth flotation followed by repeated etching in a dilute

HF-$HNO_3$ mixture.

    Chemical preparation for $^{10}$Be measurements took place in laboratories at Stanford University and the Center for Accelerator Mass Spectrometry, Lawrence Livermore National Laboratory (LLNL-CAMS). We spiked purified quartz separates with a $^{9}$Be carrier, dissolved the spiked sample in concentrated HF, purified Be by evaporation of $SiF_4$, column separation, and hydroxide precipitation, then measured $^{10}$Be/$^{9}$Be ratios using the 10 MV tandem accelerator at LLNL-CAMS. Total process and carrier

blanks were 87000 ± 12000 atoms $^{10}$Be, accounting for 1-15% of total atoms measured for most samples, but up to ∼50% in some cases. $^{10}$Be concentrations are shown in Table 1, and complete analytical data and calculations appear in Tables S1 and S2 of the supplemental material.

    Cl extraction from feldspar for $^{36}$Cl measurements took place at the Cosmogenic Nuclide Laboratory of the University of Washington. We used a double-isotope-dilution method for simultaneous measurement of $^{36}$Cl and total Cl concentrations.

We spiked 10-18 g aliquots of K-feldspar with 1.3 mg of isotopically-enriched Cl having $^{35}$Cl/$^{37}$Cl = 1.08. We then dissolved the spiked sample in a concentrated HF-$HNO_3$ mixture, separated Cl by AgCl precipitation, removed trace S by redissolution of AgCl and precipitation of $BaSO_4$, and recovered Cl by a second AgCl precipitation. Finally, we measured $^{36}$Cl/$^{37}$Cl and





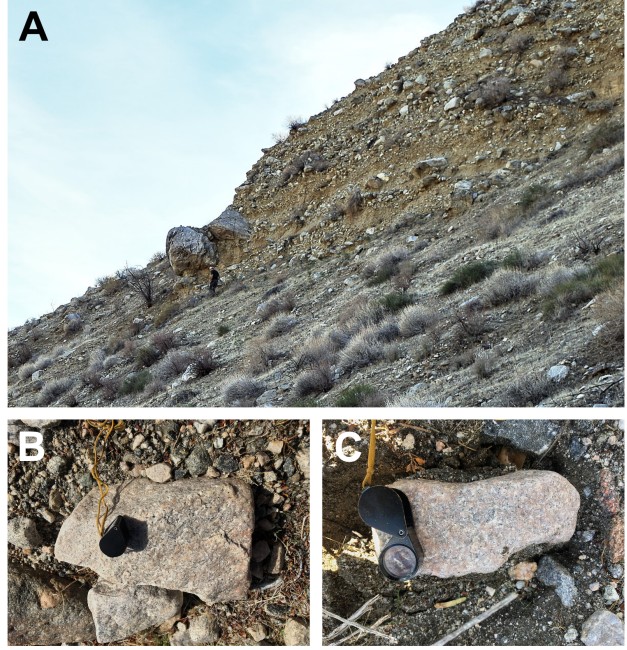

**Figure 4.** A, upper Qo at site MCP-7. Samples were collected by excavating into base of cliff face. Geologist at base of cliff for scale. B and C, representative granodiorite clasts from this site. Hand lens is 1.5 cm in diameter.

$^{35}$Cl/$^{37}$Cl ratios at LLNL-CAMS. Trace Cl present in reagent HF used for sample dissolution required us to measure both $^{36}$Cl and total Cl blanks. We found that the HF contained $0.61 \pm 0.25$ ppm Cl, which accounted for 30-60% of total measured Cl in some samples. Total process blanks contained $122,000 \pm 10000$ atoms $^{36}$Cl (mean and standard deviation of three measurements), which comprised 5-15% of total atoms measured for most samples, but up to 40% in one case. $^{36}$Cl concentrations are shown in Table 1, and complete analytical data and calculations appear in Table S3 of the supplemental material.

For measurement of major and trace element concentrations in bulk rock and target mineral separates, we supplied aliquots of crushed whole rock and clean K-feldspar separate to SGS (www.sgs.com) for measurement of major element composition by XRF (SGS "XRF76V" protocol) and concentrations of trace elements relevant to thermal neutron transport by ICP-MS ("ICM90A" protocol). Results are shown in Tables 2 and 3.

## 4 Production ratio and burial age calculations, results, and discussion

Our overall procedure for calculating burial ages is as follows. First, we calculate the expected concentration of radiogenic, that is, non-cosmogenic, $^{36}$Cl in K-feldspar separates and subtract it from total measured $^{36}$Cl to yield an estimate of the cosmogenic $^{36}$Cl concentration. Second, we compute the $^{36}$Cl/$^{10}$Be production ratio for both surface and subsurface samples using published independently calibrated estimates for reference production rates and interaction cross-sections. We then compare





these estimates with measured $^{36}$Cl/$^{10}$Be ratios in surface samples as a means of validating the production ratio estimates, and, finally, use this information to compute burial ages for subsurface samples. In general, in apportioning measured $^{36}$Cl concentrations to radiogenic and cosmogenic sources and in calculating $^{36}$Cl production rates, we follow Alfimov and Ivy-Ochs (2009), with exceptions as noted below.

## 4.1 Correction for radiogenic $^{36}$Cl

Radiogenic $^{36}$Cl arises when thermal neutrons produced by decay of natural U and Th present in rocks are captured via the reaction $^{35}$Cl(n,$\gamma$)$^{36}$Cl. Thus, when natural U/Th and Cl are present in a mineral, it is necessary to compute the $^{36}$Cl concentration resulting from this reaction and subtract it from total measured $^{36}$Cl to yield an estimate of cosmogenic $^{36}$Cl. We computed neutron yields from U and Th as well as resulting epithermal and thermal neutron fluxes using the procedure and elemental parameters given in Alfimov and Ivy-Ochs (2009). However, instead of considering only thermal neutron capture production from radiogenic neutrons as in Alfimov and Ivy-Ochs (2009), we compute both thermal and epithermal neutron capture production separately, using steady-state solutions of the two-stage neutron diffusion model described in Phillips et al. (2001). For this calculation, and also in our calculations of secondary cosmogenic thermal neutron capture production discussed below, neutron transport calculations for each sample use the bulk major and trace element concentrations measured for that sample and assume zero water content. We also assume that the radiogenic $^{36}$Cl concentration in feldspar separates has reached steady state with respect to radiogenic neutron production, which is equivalent to assuming that the bulk chemistry and U/Th concentrations of material surrounding the sample have not changed for several half-lives of $^{36}$Cl. This is not strictly true, because we know that clasts sampled for burial dating have been transported during the last several hundred thousand years from an eroding source area, where the surrounding material was presumably bedrock with the same composition as the clast, to an alluvial deposit where the surrounding material has mixed lithology. Although much of the alluvial sediment is composed of broadly granitoid lithologies and we would therefore expect the bulk chemistry of the sediment to be similar to that of granitoid clasts, we cannot verify this assumption, and it may contribute significant uncertainty to radiogenic $^{36}$Cl estimates.

Table 1 shows calculated radiogenic $^{36}$Cl concentrations and resulting implied concentrations of cosmogenic $^{36}$Cl. Although U and Th concentrations in bulk rock are generally low and Cl concentrations in feldspar separates are very low (mostly 7-12 ppm, with one outlier having 40 ppm; see Table 1), total measured $^{36}$Cl concentrations are also very low in many samples. Thus, although radiogenic $^{36}$Cl represents less than a 15% correction to total measured $^{36}$Cl for most samples, it makes up nearly 35% of measured $^{36}$Cl for some samples with the lowest total $^{36}$Cl concentrations, so the uncertainty we assign to calculated radiogenic $^{36}$Cl is important. It is unclear how to estimate this uncertainty, which is no less than measurement uncertainties of order 10% in U, Th, and Cl concentrations, but could also be much larger if the steady-state assumption discussed above proved to be incorrect. Lacking additional information, we assign a 25% uncertainty to radiogenic $^{36}$Cl estimates.

## 4.2 Production rate and production ratio calculations

Computing $^{10}$Be and $^{36}$Cl production rates and thus the $^{36}$Cl/$^{10}$Be production ratio requires assuming an elevation at which nuclide production prior to transport and burial took place. The fraction of total production attributable to high-energy neutron



spallation and muon interactions is different for $^{10}$Be and $^{36}$Cl, and these production pathways scale differently with elevation, so the assumed elevation has a small effect on the estimated production ratio. Only one of our samples (MoCan) was collected from a bedrock outcrop whose elevation is known; all others are clasts from fluvial sediments. For well-mixed samples of fluvial sand commonly used for erosion-rate estimation and burial-dating applications, one can assume that the sample represents a

mixture of sediment from uniform erosion of all parts of the watershed, and thus the sample experienced the mean production rate in the watershed before transport and burial (e.g., Bierman and Steig, 1996). In contrast, in this study we are analyzing individual clasts that were derived from a single point in their source watershed, and we have no way of determining where that point is except to observe that it must be somewhere in the area of granodiorite outcrop within the watershed. Thus, we assumed that clast samples originated at the mean elevation of portions of each watershed mapped as granodiorite, which

are 2100 m, 1000 m, and 1600 m for Mission Creek, Morongo Valley Canyon, and Little Morongo Canyon, respectively. As discussed above, the provenance of upper Qd sediments at site MCP-8 indicate only that these sediments must be derived from somewhere to the east of the field area; lacking further information we assume clasts from this site originated at the mean elevation of granite outcrop in the Little Morongo Canyon watershed. These assumptions are somewhat speculative, and clast samples could have originated from a wide range of elevations. However, even if incorrectly estimating elevations leads us to

incorrectly estimate the magnitude of nuclide production rates at sample source locations, the production ratio is very weakly sensitive to elevation, so these assumptions do not have a significant effect on production ration and burial age calculations.

For spallogenic production of $^{10}$Be in quartz as well as $^{36}$Cl production from spallation on Ca and K, we used the production rate scaling method of Stone (2000) with corresponding reference production rates of 4.01 atoms $(\mathrm{g\ quartz})^{-1}\ \mathrm{yr}^{-1}$ for $^{10}$Be, 151 atoms $(\mathrm{g\ K})^{-1}\ \mathrm{yr}^{-1}$ for $^{36}$Cl production from K, and 52 atoms $(\mathrm{g\ Ca})^{-1}\ \mathrm{yr}^{-1}$ for $^{36}$Cl production from Ca (Borchers

et al., 2016). Fe and Ti concentrations in feldspar separates are low and we assumed that $^{36}$Cl production from these elements is negligible. We assumed an effective attenuation length for spallogenic production of 150 $\mathrm{g\ cm}^{-2}$. For $^{10}$Be production by muon interactions, we used the "Model 1A" code and calibrated cross-sections from Balco (2017).

For thermal and epithermal neutron capture production of $^{36}$Cl, we use the method and physical parameters in Alfimov and Ivy-Ochs (2009), with a reference value for fast neutron production in the atmosphere of 696 $\mathrm{n\ g}^{-1}\ \mathrm{yr}^{-1}$ (Marrero et al.,

2016), again scaled for latitude and elevation according to Stone (2000). As in the calculation of radiogenic $^{36}$Cl, we assume that the bulk rock major and trace element concentration measured in each individual clast is applicable to the source area of that clast, and we assume zero water content. For K-feldspar separates in this study, nearly all $^{36}$Cl production is due to K spallation. Low total Cl concentrations result in minimal neutron capture production: $^{36}$Cl production from epithermal and thermal neutron capture is less than 3% of total production for all samples except for MC-P7-1, for which it is 6% (Figure 5).

Thus, assumptions involved in computing neutron capture production, for example the assumption of zero water content, are minimally important for these samples. For muon-induced production of $^{36}$Cl from Ca and K, we use the "Model 1A" code of Balco (2017) with negative muon capture yields calculated according to Alfimov and Ivy-Ochs (2009) and using values for the capture probability $f^*$ from Heisinger et al. (2002a), a cross-section for fast muon production from Ca from Heisinger et al. (2002b), and a cross-section for fast muon production from K from Marrero et al. (2016). Finally, we compute thermal

neutron capture production due to muon-induced neutrons according to Alfimov and Ivy-Ochs (2009). As discussed in detail




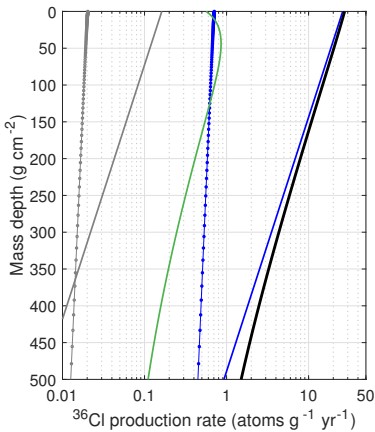
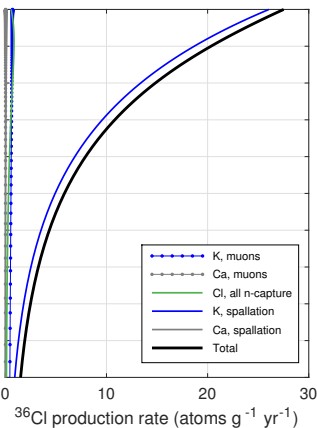

**Figure 5.** Typical $^{36}$Cl production profile for K-feldspar separates prepared from granodiorite. The data are the same in both panels; left-hand panel has logarithmic x-axis to clearly show minor production pathways. This is calculated for the location and chemical composition of surface bedrock tor sample MoCan-C. Near-surface production is dominantly spallogenic from K; Ca spallation, neutron capture on Cl, and muon interactions are minimal by comparison.

by Alfimov and Ivy-Ochs (2009), it is unclear how accurate these estimates of muon-induced production are, in particular the estimate of fast muon production from K. However, in this work we are concerned mainly with the $^{36}$Cl/$^{10}$Be production ratio in the upper several hundred $\mathrm{g\ cm^{-2}}$ below the surface, so inaccuracies in muon production rate estimates are very small in comparison to total production, and we have not considered this issue further.

Figure 6 shows calculated $^{36}$Cl and $^{10}$Be production rates and $^{36}$Cl/$^{10}$Be production ratios for both surface samples and buried clasts. Differences in $^{36}$Cl production rates among K-feldspar separates and therefore the ratio of $^{36}$Cl production in feldspar to $^{10}$Be production in quartz are primarily the result of variable K concentration in feldspar separates. As nearly all production in K-feldspar separates is from K spallation, source elevation has a minimal effect on the near-surface production ratio. However, the relatively large cross-sections estimated for muon production on K predict that the $^{36}$Cl-in-feldspar/$^{10}$Be-
in-quartz production ratio increases significantly in the shallow subsurface where a significant fraction of $^{36}$Cl production is due to muon interactions. Significant thermal neutron capture production from Cl can also increase the production ratio in the subsurface; this is evident in Figure 6 for sample MC-P7-1, which has a higher Cl concentration than the others.

### 4.3  Comparison of calculated and measured ratios in surface samples

Figure 7 compares measured $^{36}$Cl and $^{10}$Be concentrations in one bedrock surface sample and three clasts from modern
fluvial systems with predicted $^{36}$Cl-in-feldspar/$^{10}$Be-in-quartz ratios. Because the predicted production ratio is different in each sample, we first calculate normalized concentrations by dividing measured nuclide concentrations by calculated surface production rates, and compare them with expected concentrations for (i) simple exposure at zero erosion and (ii) steady-state erosion. If these samples did, in fact, arise from steady erosion of granodiorite bedrock in source watersheds and we have





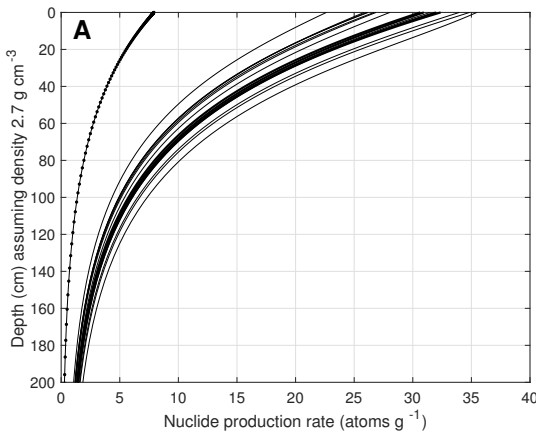
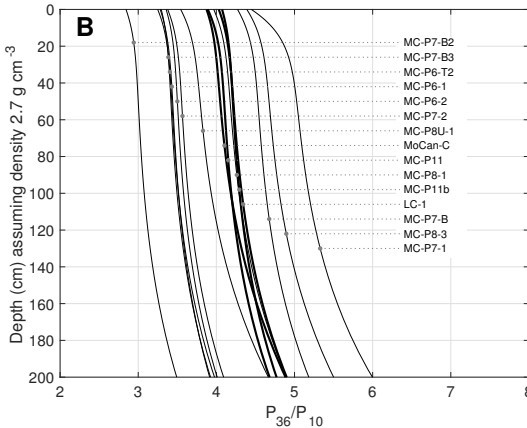

**Figure 6.** Predicted $^{36}$Cl and $^{10}$Be production rates and $^{36}$Cl-in-feldspar/$^{10}$Be-in-quartz production ratios for samples in this study. Left panel, calculated $^{10}$Be-in-quartz (dotted line) and $^{36}$Cl-in-feldspar (solid lines) production rates for all samples. In this panel, we assumed a common elevation (1000 m) for all samples to highlight the variation in $^{36}$Cl production rates among samples due to varying K concentrations in feldspar separates. Right panel, corresponding predicted $^{36}$Cl/$^{10}$Be production ratios in the shallow subsurface. In this panel, elevations used in the calculation vary with sample source watershed, as described in the text, although the effect of elevation change on the near-surface production ratio is small. In both panels, bold lines are surface samples (MoCan-C, MCP-11, MCP-11A, LC-1) and lighter lines are buried clast samples.

correctly calculated production ratios for the surface samples, normalized measured concentrations should coincide with the steady-state erosion line.

The observed $^{36}$Cl/$^{10}$Be ratio in the surface bedrock tor sample (MoCan-C; rightmost ellipse on Fig. 7), which has relatively high $^{36}$Cl and $^{10}$Be concentrations, is exactly consistent with our production rate calculations and the assumption of steady

erosion. The implied rock surface erosion rate, computed from the $^{10}$Be concentration using version 3 of the online erosion rate calculator described by Balco et al. (2008) and subsequently updated, is $63.3 \pm 5.6$ m Myr$^{-1}$, which is typical for low-relief areas in this region (Binnie et al., 2010). This sample represents the most reliable test of our production rate calculations that we have available, because it has relatively high nuclide concentrations that can be measured with good precision, and in addition it is the only one whose location and elevation during exposure are known. Thus, the agreement between predictions

and measurements for this sample indicates that our calculated surface production ratios for this lithology, which primarily depend on the absolute calibration of the $^{10}$Be and $^{36}$Cl-from-K-spallation production rates, are accurate.

On the other hand, observed $^{36}$Cl/$^{10}$Be ratios in surface fluvial clasts are less precisely measured because nuclide concentrations are an order of magnitude lower, so counting uncertainties and blank subtraction uncertainties are correspondingly larger. Although the individual measurements are not distinguishable from predicted ratios at high confidence, overall they are

15 systematically ∼15% higher than predicted by our production rate calculations. It is difficult to explain this disagreement by an underestimate of radiogenic $^{36}$Cl, because radiogenic $^{36}$Cl makes up less than ∼6% of total $^{36}$Cl in these samples, and it





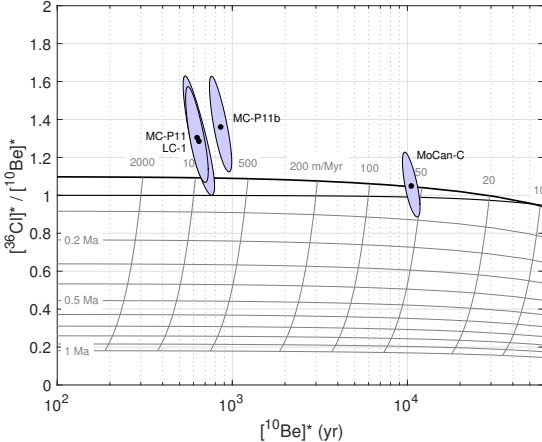

**Figure 7.** Normalized $^{36}$Cl/$^{10}$Be diagram comparing normalized nuclide concentrations (measured cosmogenic $^{36}$Cl and $^{10}$Be concentrations divided by calculated surface production rates at estimated sample source elevations) in clasts from modern fluvial sytems (lower-concentration data: LC-1, MCP-11, MCP-11a) and one bedrock tor (MoCan-C) with predictions for simple exposure at zero erosion (lower dark line) and steady erosion (upper dark line). The simple exposure and steady erosion lines are drawn for the average chemical composition of all samples in this study. Note that the steady erosion line lies above the simple exposure line for high erosion rates in this plot because of the prediction that the subsurface $^{36}$Cl-in-feldspar/$^{10}$Be-in-quartz production ratio is higher than the surface ratio due to relatively high muon production rates on K. The ellipses are 68% confidence regions that include both measurement uncertainty for nuclide concentrations and 7% uncertainties in $^{36}$Cl and $^{10}$Be production rates. Light gray lines are contours of erosion rate and burial time; burial age contours are drawn assuming that the initial condition is steady erosion.

cannot be explained by our assumption of zero water content, because assuming a nonzero water content would worsen the agreement. A possible explanation comes from the observation that nuclide concentrations in these clasts are unexpectedly low. If we assume that exposure took place at mean elevations of granite outcrop in watersheds as discussed above, these concentrations are equivalent to 600-1300 years of surface exposure or steady erosion at 600-1000 m Myr$^{-1}$. Although erosion rates this high are common in steeper areas of the San Bernardino Mountains (Binnie et al., 2010), nuclide concentrations in these clasts are lower by a factor of 2 than observed in bulk sediment from the same watersheds (Fosdick and Blisniuk, 2018). Lower-than-expected concentrations and higher-than-predicted ratios suggest that a possible explanation for this mismatch is that these clasts were not exposed at the surface in sediment source areas, but instead accumulated most of their nuclide concentration in the shallow subsurface rather than at the surface. As shown in Figure 6, $^{36}$Cl/$^{10}$Be production ratios 15% higher than the surface production ratio are predicted to be characteristic of depths between 1-1.8 m below the surface. Thus, lower nuclide concentrations in clasts than in bulk sediment and higher-than-expected $^{36}$Cl/$^{10}$Be ratios could potentially both be explained if large clasts in fluvial systems are commonly delivered to the channel by episodic shallow landsliding and not by continuous surface erosion, as has been proposed elsewhere to explain variation in $^{10}$Be concentrations with sediment grain size (e.g., Brown et al., 1995). On the other hand, a weakness of this explanation is that production rates are also much lower





below the surface, so this explanation would require slower erosion rates in clast source areas than expected based on $^{10}$Be concentrations in bulk stream sediments. Regardless, the observation that measured $^{36}$Cl/$^{10}$Be ratios in clasts in the modern fluvial system may be higher than predicted, either because of a subsurface origin for the clasts or unrecognized errors in estimates of production rates or nucleogenic $^{36}$Cl concentrations, creates a problem for burial-dating of subsurface clasts observed

to have low nuclide concentrations, because it makes it unclear how to choose a surface production ratio for the burial age calculation in this case.

To summarize, agreement between the calculated and observed $^{36}$Cl/$^{10}$Be ratio for one surface bedrock sample with relatively high nuclide concentration validates our calculations of the surface production ratio from rock and target mineral chemical compositions, but on the other hand highlights that for fluvial clasts with very low nuclide concentrations, the fact

that clast exposure prior to detachment and transport may have taken place in the shallow subsurface rather than at the surface creates an obstacle to accurately estimating the production ratio for burial dating.

### 4.4   Nuclide concentrations in subsurface samples

Figure 8 shows normalized $^{36}$Cl and $^{10}$Be concentrations in alluvial clasts collected for burial-dating. With one exception (MC-P7-B), nuclide concentrations are extremely low, lower even than in clasts collected from modern fluvial channels (shown

in Fig. 7). Thus, if none of the measured nuclide concentrations represent post-burial production, these clasts most likely originated from source areas with erosion rates greater than 1 km Myr$^{-1}$. If some nonzero fraction of the current nuclide inventory was produced during burial or re-exhumation of samples, which is nearly certain to be the case, then erosion rates could be higher. This is possible, as basin-scale erosion rates in the San Bernardino Mountains are observed to be as high as 2.7 km Myr$^{-1}$ (Binnie et al., 2010). However, extremely low nuclide concentrations cause a number of difficulties with using these

data to compute burial ages. First, measurement uncertainties for both $^{10}$Be and $^{36}$Cl are large for these nuclide concentrations, and a significant correction for radiogenic $^{36}$Cl for some samples further increases the uncertainty in the estimate of cosmogenic $^{36}$Cl. Second, as discussed above, rapid erosion rates and low concentrations imply that much of the observed $^{36}$Cl and $^{10}$Be may have been produced in the shallow subsurface rather than at the surface, which makes it difficult to estimate the initial $^{36}$Cl/$^{10}$Be ratio in clasts at the time of burial. Third, nuclide concentrations are low enough that, even though the clasts were

collected from shielded outcrops, nearly all the observed nuclide concentrations could have been produced after burial, either by muon-induced production when buried within the alluvial deposit or near the surface during exhumation by recent incision of the sediment pile. For example, MC-P8U-1 has 5000 atoms g$^{-1}$ $^{10}$Be, which could be produced in only 1500 yr of exposure at the present location and depth where we collected the sample.

In keeping with the hypothesis that a substantial fraction of the measured nuclide concentrations in most of the buried clasts

is likely the result of post-burial production, $^{36}$Cl/$^{10}$Be ratios for low-nuclide-concentration buried samples, shown on Figure 8, are scattered around the production ratio and do not show evidence of significant burial. In fact, although uncertainties are large enough that observed and predicted ratios cannot be distinguished at high confidence for most samples (the ellipses on Fig. 8 are 68% confidence regions, so nearly all data from the low-nuclide-concentration clasts would overlap the simple exposure region at 95% confidence), some observed ratios are higher than the surface production ratio, and, in general, the





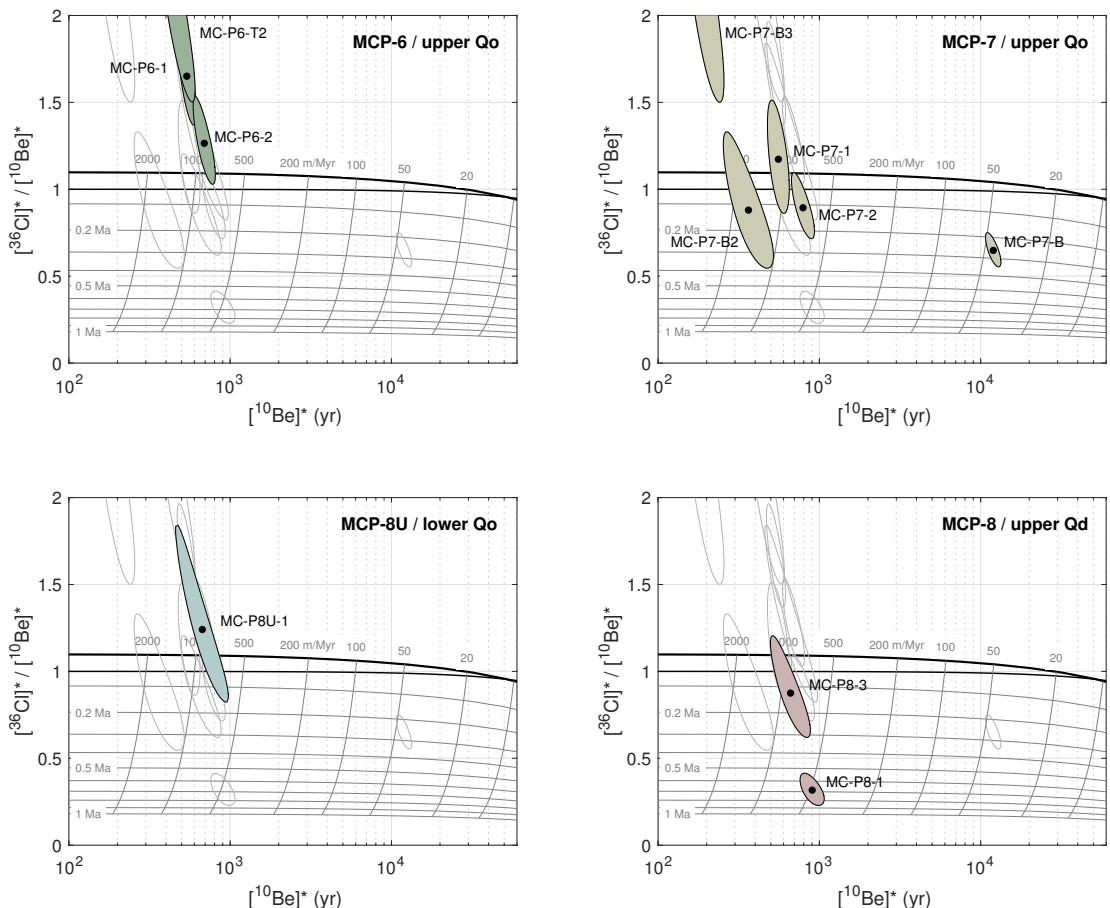

**Figure 8.** Normalized $^{36}$Cl/$^{10}$Be diagrams comparing normalized nuclide concentrations in buried clasts with predictions for simple exposure at zero erosion (lower dark line) and steady erosion (upper dark line). Figure elements are as described in Figure 7. Each panel highlights data from a single stratigraphic unit as filled ellipses, but all ellipses are shown as outlines in all panels for ease of comparison between stratigraphic units.





array of data shows an inverse relationship between nuclide concentrations and normalized $^{36}$Cl/$^{10}$Be ratio. Overall, these observations are best explained if the majority of $^{10}$Be and $^{36}$Cl in these clasts was produced in the subsurface during burial or re-exhumation of the outcrop we sampled. For example, all the measurements from unit MCP-6 (upper left panel of Fig. 8 nuclide concentrations in these three clasts are effectively indistinguishable) could be explained by 0.2 Ma burial at a depth

of 1000 g cm$^{-2}$. Finally, because nuclide concentrations show little variation among clasts at each site (with one exception, described in the next section), it is not possible to use an isochron method to account for postdepositional production. Thus, it is not feasible to compute burial ages for the majority of our buried clast samples.

## 4.5 Burial ages for some subsurface samples

In contrast to the majority of buried clast samples that display low nuclide concentrations and $^{36}$Cl/$^{10}$Be ratios characteristic

of post-burial nuclide production, one sample (MC-P7-B; upper right panel of Fig. 8) has much higher nuclide concentrations, indicating an origin from a lower-erosion-rate site, and a $^{36}$Cl/$^{10}$Be ratio well below the production ratio, indicating extended burial. In fact, nuclide concentrations in this sample are higher than observed in the surface bedrock sample (MoCan-C), meaning that this clast most likely originated from a relatively high-elevation site with an erosion rate similar to or lower than the surface bedrock site. High-elevation, low-relief surfaces with low erosion rates do exist in the San Bernardino Mountains

(Binnie et al., 2010), so this is possible. The $^{10}$Be concentration in this sample is a factor of 25 higher than the average concentration in other buried clasts from the MCP-7 site; if we take the lower-concentration samples to represent an upper limit on post-burial nuclide production, this limits post-burial production to less than 4% of the total measured $^{10}$Be concentration in this sample. In addition, high nuclide concentrations likely indicate that it originated from steady erosion at the surface, so our calculated surface production ratio is likely accurate for a burial-dating calculation. Thus, we can compute a $^{36}$Cl/$^{10}$Be

burial age for this sample by assuming a two-stage exposure history including initial equilibrium with steady erosion followed by a single period of burial at infinite depth.

    If we assume zero post-burial production, this yields a burial age of 260 ± 17 (56) ka for upper Qo at site MCP-7. These uncertainties are internal (including measurement uncertainties only) and external (in parentheses, also including uncertainties in production rates and decay constants) uncertainties derived from a Monte Carlo simulation. If we observe that the clasts with

the lowest nuclide concentrations at this site (MC-P7-B2 and MC-P7-B3) provide an upper limit on total post-burial nuclide production and calculate the burial age on that basis, we obtain a nearly identical burial age estimate of 268 ± 20 (57) ka. Note that this procedure is equivalent to computing an isochron age (effectively a two-point isochron) from all clasts at the MCP-7 site. Finally, following the discussion above, we consider how sensitive this burial age is to the assumed elevation of initial exposure. If we assume that this clast originated at 1000 m instead of 2100 m (see discussion above), the resulting burial age

would be 240 ka, which highlights that even though assumed source elevations for clasts are poorly constrained, the burial age calculation is not very sensitive to this assumption.

    A second buried sample, MC-P8-1 (lower right panel of Figure 8) has very low nuclide concentrations but a $^{36}$Cl/$^{10}$Be ratio that is significantly below the production ratio, so it is theoretically possible to compute a burial age for this sample. Again assuming zero postdepositional production, this yields an apparent burial age of 678 ± 99 (123) ka. This is a stratigraphically



possible age for upper Qd, but is not consistent with $^{36}$Cl and $^{10}$Be concentrations in MC-P8-3, the other sample from this site, whose $^{36}$Cl/$^{10}$Be ratio is indistinguishable from the production ratio (Figure 8). In addition, the two samples would form an isochron with an unphysical negative slope. On the other hand, both samples have very low nuclide concentrations, so the disagreement could be the result of unrecognized errors in the concentration measurements or the radiogenic $^{36}$Cl calculations,

in which case both samples might be consistent with a burial age for upper Qd near 0.5 Ma. Overall, because of the difficulty of interpreting data for clasts with very low nuclide concentrations, burial age estimates for upper Qd from these two samples are not credible.

## 5    Summary discussion and conclusions

We have successfully applied the $^{36}$Cl/$^{10}$Be nuclide pair for burial dating of alluvial sediments that are younger than can

usually be accurately dated with the more commonly used $^{26}$Al/$^{10}$Be pair. For the minority of samples in this study that arise from relatively low-erosion-rate environments and therefore have relatively high nuclide concentrations, we use a surface bedrock sample to show that production rate calculations based on sample composition and independently calibrated parameters correctly predict the $^{36}$Cl-in-feldspar/$^{10}$Be-in-quartz production ratio, at least for K-rich, Cl-poor feldspars characteristic of typical granitoid rocks. We then use this information to compute the burial age of one buried sample. The resulting burial

age has nominal measurement precision better than could likely be attained with the $^{26}$Al/$^{10}$Be pair in this age range (Figure 1), and is consistent with preexisting age constraints as well as fault slip reconstructions. Geomorphic relations and provenance data indicate that upper Qo has been displaced along the Mission Creek strand of the San Andreas Fault relative to its source area by at least 4 km (Fosdick and Blisniuk, 2018). Also, incised into upper Qo is an abandoned channel of Mission Creek ("former Mission Creek" in Figure 2) that is offset by 2.5 km (Kendrick et al., 2015). The $260 \pm 57$ ka age for upper Qo at

site MCP-7 is a maximum age for abandonment of the buried channel, which implies a strict minimum slip rate of $10 \pm 2$ mm yr$^{-1}$ over 260 ka. Likewise, in conjunction with the minimum of 4 km of offset for Qo inferred from the provenance analysis, this age implies a minimum slip rate of $16 \pm 3$ mm yr$^{-1}$ over 260 ka. These minimum constraints, combined with maximum constraints of 20-30 mm yr$^{-1}$ (Fosdick and Blisniuk, 2018), are consistent with the conclusions of Fosdick and Blisniuk (2018) that the Mission Creek strand has accommodated a significant fraction of the total right-lateral slip on the

San Andreas fault system during the late Quaternary. Overall, although it is difficult to evaluate the accuracy of a single burial age without internal reliability criteria derived from comparison of multiple independent ages, our results are consistent with available observations and we conclude that upper Qo alluvium at the MCP-7 level is 260 ka.

On the other hand, the majority of the samples we analyzed in this study highlighted difficulties in burial dating when samples originate from rapidly eroding landscapes and have commensurately low nuclide concentrations, including (i) measurement

uncertainty generally; (ii) uncertainties in estimating nucleogenic $^{36}$Cl; and (iii) the fact that even though post-burial nuclide concentrations are very low in rapidly accumulating fault-proximal alluvium, pre-burial concentrations are even lower. As the extraordinary erosion rates of San Bernardino Mountains and neighboring ranges are so well known as to be emblematic of catastrophic erosion and sediment transport (McPhee, 1988a, b), perhaps we should not have been surprised to discover that

(c) Author(s) 2019. CC BY 4.0 License.





most fluvial clasts derived from these mountains have very low $^{36}$Cl and $^{10}$Be concentrations. In general, cosmogenic-nuclide burial dating is most effective when samples have high initial nuclide concentrations due to long residence at low erosion rates in their source areas, and low post-burial nuclide concentrations due to rapid burial in sedimentary deposits. Although landscapes with high sediment yields are unfavorable for the first of these, they should be favorable for the second. In this

study, we found that the balance of these two effects lay once on the winning side, but mostly on the losing side. The challenge in future applications of $^{36}$Cl/$^{10}$Be burial dating in this system is that of locating the few rare clasts that have originated from low-erosion-rate parts of the landscape.

*Code and data availability.* Spreadsheets containing complete data reduction calculations for $^{36}$Cl and $^{10}$Be measurements, as well as MAT-LAB code used for all calculations and to generate Figures 5-8, can be downloaded from this address:

`http://hess.ess.washington.edu/repository/bd36/`

*Competing interests.* Balco is an editor of *Geochronology.*

*Acknowledgements.* We thank John Stone for making the UW Cosmogenic Nuclide Laboratory available for this work and for assistance with Cl extraction chemistry. In addition, we thank Christopher and Randee Johnson, Louis Wersen, and Jesse Waco for field assistance, Katherine Guns for laboratory assistance, and Katherine Scharer and John Matti for helpful discussions. Funding for this work was provided

by Southern California Earthquake Center (SCEC) grants 14103 and 15127 and by the Ann and Gordon Getty Foundation. Portions of the work were performed under the auspices of the U.S. Department of Energy by Lawrence Livermore National Laboratory under Contract DE-AC52-07NA27344. This is LLNL-JRNL-759420.



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





**Table 1.** Sample locations, $^{10}$Be concentrations in quartz and $^{36}$Cl concentrations in feldspar separates. $^{10}$Be concentrations are normalized to the '07KNSTD' standardization of Nishiizumi et al. (2007), and $^{36}$Cl concentrations to standards described by Sharma et al. (1990). Detailed analytical data appear in the supplement.

| Sample | Latitude (DD) | Longitude (DD) | Elevation (m) | $[^{10}$Be] (atoms g$^{-1}$) | [Cl] (ppm) | Measured total [$^{36}$Cl] (atoms g$^{-1}$) | Estimated radiogenic [$^{36}$Cl] (atoms g$^{-1}$) | Percent radiogenic | Cosmogenic [$^{36}$Cl] (atoms g-1) | Predicted $^{36}$Cl/$^{10}$Be production ratio | Measured $^{36}$Cl/$^{10}$Be ratio |
|---|---|---|---|---|---|---|---|---|---|---|---|
| Clasts in modern river channels crossing Mission Creek Fault | | | | | | | | | | | |
| LC-1 | 33.972 | -116.443 | 489 | 7836 ± 947 | 8.8 ± 1.0 | 42862 ± 2316 | 2269 ± 567 | 5 | 40593 ± 1987 | 4.03 | 5.18 ± 0.68 |
| MC-P11 | 34.020 | -116.633 | 766 | 10993 ± 768 | 10.4 ± 1.1 | 59324 ± 2531 | 3528 ± 882 | 6 | 55796 ± 2312 | 3.89 | 5.08 ± 0.41 |
| MC-P11b | 34.020 | -116.633 | 766 | 14993 ± 1087 | 6.9 ± 1.1 | 87571 ± 2884 | 4483 ± 1121 | 5 | 83088 ± 2708 | 4.07 | 5.54 ± 0.44 |
| Surface bedrock exposure north of Mission Creek Fault | | | | | | | | | | | |
| MoCan-C | 34.0436 | -116.5940 | 856 | 74714 ± 2956 | 10.0 ± 1.0 | 308066 ± 11104 | 4288 ± 1072 | 1 | 303778 ± 11156 | 3.87 | 4.07 ± 0.22 |
| Site MCP-6, upper Qo | | | | | | | | | | | |
| MC-P6-1 | 34.024 | -116.645 | 959 | 9417 ± 555 | 7.0 ± 1.0 | 51440 ± 2161 | 440 ± 110 | 1 | 51000 ± 1783 | 3.28 | 5.42 ± 0.37 |
| MC-P6-2 | " | " | " | 12083 ± 1004 | 7.0 ± 1.0 | 55371 ± 2324 | 4075 ± 1019 | 7 | 51296 ± 2224 | 3.36 | 4.25 ± 0.4 |
| MC-P6-T2 | " | " | " | 7758 ± 1605 | 10.6 ± 1.1 | 65079 ± 2631 | 10937 ± 2734 | 17 | 54142 ± 3794 | 3.25 | 6.98 ± 1.52 |
| Site MCP-7, upper Qo | | | | | | | | | | | |
| MC-P7-1 | 34.023 | -116.641 | 920 | 9736 ± 826 | 39.5 ± 1.2 | 78878 ± 2415 | 28098 ± 7025 | 36 | 50780 ± 7344 | 4.45 | 5.22 ± 0.87 |
| MC-P7-2 | " | " | " | 13841 ± 1264 | 9.6 ± 1.0 | 49320 ± 2130 | 7551 ± 1888 | 15 | 41769 ± 2571 | 3.38 | 3.02 ± 0.33 |
| MC-P7-B | 34.021 | -116.639 | 886 | 209325 ± 5914 | 12.7 ± 1.0 | 584213 ± 8436 | 4875 ± 1219 | 1 | 579338 ± 8370 | 4.27 | 2.77 ± 0.09 |
| MC-P7-B2 | " | " | " | 6353 ± 1556 | 9.0 ± 1.1 | 24024 ± 1459 | 8125 ± 2031 | 34 | 15899 ± 2501 | 2.84 | 2.5 ± 0.73 |
| MC-P7-B3 | " | " | " | 3212 ± 712 | 6.9 ± 1.6 | 28263 ± 2315 | 4427 ± 1107 | 16 | 23836 ± 2566 | 3.29 | 7.42 ± 1.83 |
| Site MCP-8U, lower Qo | | | | | | | | | | | |
| MC-P8U-1 | 34.021 | -116.636 | 804 | 4932 ± 817 | 12.4 ± 1.1 | 24506 ± 1451 | 949 ± 237 | 4 | 23557 ± 1470 | 3.54 | 4.4 ± 1.15 |
| Site MCP-8, upper Qd | | | | | | | | | | | |
| MC-P8-1 | 34.021 | -116.636 | 801 | 10964 ± 1074 | 10.6 ± 1.2 | 17949 ± 2372 | 4176 ± 1044 | 23 | 13773 ± 2029 | 3.97 | 1.26 ± 0.22 |
| MC-P8-3 | " | " | " | 8053 ± 1494 | 15.8 ± 1.0 | 34091 ± 2616 | 3157 ± 789 | 9 | 30934 ± 2732 | 4.39 | 3.84 ± 0.79 |

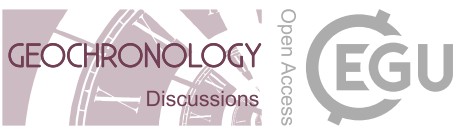

**Table 2.** Major element composition in whole rock and K-feldspar separates.

| Sample name | Oxide weight percents measured by XRF | | | | | | | | | | |
| | $SiO_2$ | $TiO_2$ | $Al_2O_3$ | $Fe_2O_3$ | MnO | MgO | CaO | $Na_2O$ | $K_2O$ | $P_2O_5$ | LOI (wt %) |
|---|---|---|---|---|---|---|---|---|---|---|---|
| **Bulk rock** | | | | | | | | | | | |
| LC-1 | 71.4 | 0.1 | 15.4 | 2.2 | 0.0 | 0.4 | 2.0 | 3.8 | 4.3 | 0.1 | 0.5 |
| MC-P11 | 78.1 | 0.0 | 12.4 | 0.5 | 0.0 | 0.1 | 1.4 | 3.8 | 2.9 | 0.0 | 0.5 |
| MC-P11b | 74.5 | 0.1 | 12.9 | 3.1 | 0.0 | 0.1 | 1.0 | 2.6 | 5.9 | 0.0 | 0.0 |
| MoCan-C | 72.2 | 0.1 | 14.1 | 2.0 | 0.0 | 0.1 | 0.5 | 2.6 | 7.9 | 0.0 | 0.3 |
| MC-P6-1 | 77.3 | 0.1 | 12.0 | 1.5 | 0.0 | 0.1 | 0.3 | 3.7 | 3.7 | 0.1 | 0.5 |
| MC-P6-2 | 73.8 | 0.1 | 14.2 | 1.9 | 0.0 | 0.3 | 1.3 | 3.5 | 4.3 | 0.1 | 0.6 |
| MC-P6-T2 | 72.6 | 0.1 | 14.1 | 2.7 | 0.0 | 0.2 | 1.1 | 3.2 | 5.4 | 0.0 | 0.2 |
| MC-P7-1 | 71.5 | 0.3 | 14.2 | 3.7 | 0.0 | 0.5 | 1.8 | 2.7 | 5.4 | 0.3 | 0.9 |
| MC-P7-2 | 73.3 | 0.1 | 13.8 | 2.4 | 0.1 | 0.3 | 1.1 | 3.3 | 4.8 | 0.0 | 0.4 |
| MC-P7-B | 79.5 | 0.1 | 11.2 | 0.8 | 0.0 | 0.1 | 0.8 | 2.3 | 5.1 | 0.0 | 0.3 |
| MC-P7-B2 | 70.5 | 0.4 | 14.5 | 3.9 | 0.1 | 0.2 | 1.7 | 3.4 | 4.3 | 0.2 | 0.9 |
| MC-P7-B3 | 66.5 | 0.5 | 15.5 | 4.8 | 0.1 | 0.8 | 2.2 | 3.6 | 3.9 | 0.2 | 1.5 |
| MC-P8-1 | 70.7 | 0.3 | 14.4 | 2.9 | 0.1 | 0.7 | 2.1 | 3.3 | 4.2 | 0.1 | 0.4 |
| MC-P8U-1 | 72.9 | 0.1 | 13.8 | 2.7 | 0.0 | 0.3 | 1.6 | 3.1 | 4.8 | 0.1 | 0.2 |
| MC-P8-3 | 73.3 | 0.1 | 13.5 | 2.6 | 0.0 | 0.2 | 1.6 | 2.3 | 5.9 | 0.0 | 0.0 |
| **K-feldspar fraction** | | | | | | | | | | | |
| LC-1 | 66.0 | 0.0 | 18.6 | 0.1 | 0.0 | 0.0 | 0.4 | 2.3 | 12.4 | 0.0 | - |
| MC-P11 | 65.3 | 0.0 | 18.5 | 0.2 | 0.0 | 0.0 | 0.4 | 2.5 | 11.9 | 0.0 | - |
| MC-P11b | 64.5 | 0.0 | 18.1 | 0.2 | 0.0 | 0.0 | 0.4 | 2.1 | 12.6 | 0.0 | - |
| MoCan-C | 64.6 | 0.0 | 17.9 | 0.9 | 0.0 | 0.0 | 0.3 | 2.5 | 11.9 | 0.0 | - |
| MC-P6-1 | 65.6 | 0.0 | 18.9 | 0.3 | 0.0 | 0.0 | 0.3 | 3.8 | 10.1 | 0.1 | - |
| MC-P6-2 | 65.2 | 0.0 | 18.8 | 0.5 | 0.0 | 0.0 | 0.8 | 3.3 | 10.2 | 0.0 | - |
| MC-P6-T2 | 64.7 | 0.0 | 19.1 | 0.8 | 0.0 | 0.0 | 1.0 | 3.7 | 9.7 | 0.0 | - |
| MC-P7-1 | 64.5 | 0.0 | 18.1 | 0.1 | 0.0 | 0.0 | 0.3 | 1.9 | 13.0 | 0.0 | - |
| MC-P7-2 | 65.3 | 0.0 | 19.4 | 0.2 | 0.0 | 0.0 | 0.8 | 3.6 | 10.2 | 0.0 | - |
| MC-P7-B | 64.9 | 0.0 | 18.2 | 0.2 | 0.0 | 0.1 | 0.2 | 1.9 | 13.1 | 0.0 | - |
| MC-P7-B2 | 66.8 | 0.0 | 17.6 | 1.3 | 0.0 | 0.0 | 1.1 | 3.5 | 8.4 | 0.0 | - |
| MC-P7-B3 | 64.9 | 0.0 | 19.4 | 1.0 | 0.0 | 0.0 | 1.1 | 3.4 | 9.9 | 0.0 | - |
| MC-P8-1 | 66.3 | 0.0 | 17.8 | 0.2 | 0.0 | 0.0 | 0.5 | 2.0 | 12.1 | 0.0 | - |
| MC-P8U-1 | 65.8 | 0.0 | 17.8 | 1.4 | 0.0 | 0.0 | 0.8 | 2.7 | 10.6 | 0.1 | - |
| MC-P8-3 | 64.7 | 0.0 | 18.0 | 0.6 | 0.0 | 0.0 | 0.2 | 1.5 | 13.4 | 0.0 | - |



**Table 3.** Whole-rock concentrations (ppm) of trace elements relevant to neutron flux calculations . bdl = below detection limit.

| Sample name | Li | B | Sm | Gd | U | Th |
| --- | --- | --- | --- | --- | --- | --- |
| LC-1 | 12 | bdl | 2.9 | 2.2 | 0.8 | 7.9 |
| MC-P11 | 120 | bdl | 2.3 | 2.0 | 1.5 | 9.5 |
| MC-P11b | 11 | bdl | 3.9 | 3.1 | 3.6 | 22.0 |
| MoCan-C | bdl | 20 | 6.1 | 4.6 | 1.1 | 22.4 |
| MC-P6-1 | bdl | bdl | 0.3 | 0.3 | 0.4 | 1.2 |
| MC-P6-2 | 32 | bdl | 3.3 | 2.9 | 1.9 | 18.9 |
| MC-P6-T2 | 22 | bdl | 5.0 | 4.1 | 2.8 | 41.7 |
| MC-P7-1 | 14 | bdl | 9.5 | 10.7 | 4.2 | 24.3 |
| MC-P7-2 | 34 | bdl | 3.5 | 3.1 | 2.5 | 28.1 |
| MC-P7-B | bdl | bdl | 3.7 | 2.9 | 1.5 | 13.5 |
| MC-P7-B2 | bdl | bdl | 3.9 | 2.4 | 2.9 | 33.9 |
| MC-P7-B3 | 15 | bdl | 4.0 | 2.6 | 2.1 | 23.9 |
| MC-P8-1 | 21 | bdl | 2.8 | 2.2 | 1.5 | 12.7 |
| MC-P8U-1 | bdl | bdl | 1.0 | 1.3 | 0.9 | 0.8 |
| MC-P8-3 | bdl | bdl | 0.7 | 0.6 | 1.4 | 5.9 |