# Peer review of "Chlorine-36 / beryllium-10 burial dating of alluvial fan sediments associated with the Mission Creek strand of the San Andreas Fault system, California, USA"

_Geochronology, 2019_

## Referee Comment (RC1) · Irene Schimmelpfennig (Referee) · 29 May 2019

General comments

This excellent paper presents a new approach to burial dating using the cosmogenic nuclide pair 10Be and 36Cl, respectively measured in quartz and K-feldspar of the same sediment clasts. The study is very well designed and presented, and represents a significant contribution to the field of quantitative geomorphology, as this novel strategy extents burial age determinations towards younger time ranges in comparison with the commonly applied 10Be-26Al nuclide pair. Although only few data could finally be used to actually conclude on the burial age of the studied sediment units

(and thus on the minimum slip rate of the related section of the San Andreas Fault), the authors used the inconclusive measurements to explore the limits and challenges of the method, which will be very helpful for future applications of paired 10Be-36Cl measurements. I only have a few mostly very minor comments and suggestions for clarifications.

Specific comments

Line 22 – page 2, line 1: maybe clarify that "sediment is exposed to the surface cosmic-ray flux during erosion" refers to processes in the source area of the material and also includes bedrock

Page 2, 7-9: It would be appropriate to be a bit more specific, i.e. to give examples of the methods that you compare the cosmogenic nuclide burial dating with and add references. Do you mean to imply here that burial dating is only useful in arid regions?

Line 22: Please clarify "short relative to the half-life...": what does this quantitatively mean and why is it important? E.g. in the case of 36Cl, should the surface exposure period be less than 300 ka? Otherwise the ratio would be dominated by the decay rate?

Page 3, line 17-18: Bierman et al., GCA (1995) attempted to use cosmogenic 36Cl produced from Cl in fluid inclusions in quartz to quantify erosion rates and exposure ages. I think it would be appropriate to cite this study.

Page 5, line 28: "dated to 0.5 – 1.1 Ma" using which method?

Page 5, Line 33 – Page 6, lines 2: I guess the samples from the three stratigraphic levels were taken along a horizontal transect and not on a vertical transect due to issues of accessibility?

Page 7 line 14: "a double-isotope-dilution method": I guess you refer to the routinely used isotope dilution technique that is described e.g. in Ivy-Ochs et al, NIMB (2004) and Desilets et al., Chemical Geology (2006)? Otherwise please clarify.

Page 8, line 7: in which SGS lab the aliquots were analyzed?

Page 9, line 6: Please note that thermal neutrons are directly produced during spontaneous fission of U, but indirectly during decay of U and Th through alpha,n-reactions (see Gosse and Phillips, 2001; therein called "nucleogenic" instead of radiogenic 36Cl, which would be probably more correct)

Page 10, lines 1-2 and 16: Please quantify the "small effect" of these different muon-related scaling differences. E.g. according to the models you use for the calculations, what are the muogenic 10Be and 36Cl contributions in the range of altitudes of the considered catchments?

Page 11, line 8: for completeness it could be added that 36Cl production from K spallation and 10Be production in quartz are assumed to scale identically.

Lines 9-11: Here again, please quantify the "significant fraction" of 36Cl produced from muons in the considered depth range. This sentence seems in contradiction with lines 1-4 of the same page, where you imply that the muon-related 36Cl production and associated inaccuracies in the considered subsurface are low enough to be insignificant. Please clarify.

Lines 15-17: It should probably be added that the predicted production ratios depend on altitude and feldspar composition and are therefore different. Which sample-specific characteristics are included in the "calculated surface production rates" that you use to normalize the nuclide concentration: scaling, but also feldspar composition, right? I guess the calculations are those in equation 1?

Page 11, line 17-Page 12, line 2: What would happen if the samples were not steadily eroded? On page 16, lines 18-19, you seem to imply that if the sample didn't originate from a steadily eroding surface, the burial age could not be determined – probably due to the varying production rate ratio? Is this issue unique to the 36Cl-10Be pair?

Page 12, lines 9-11: this seems like a circular argument: you use the comparison of

the observed to calculated ratio to check whether or not the sample is at steady state erosion (lines 3-5), and here you use the same comparison to conclude that the applied reference production rates are accurate. Please clarify.

Fig. 7, caption: Regarding the sentence that starts with "Note that the steady erosion...", please specify "high erosion rates". According to the diagram, the simple exposure line lies below the steady erosion line from the left-most erosion rate ($\sim$15 m/Ma) on, so over the whole spectrum of considered erosion rates.

Page 16, lines 3-5: Please clarify how you determined 0.2 Ma burial at a depth of 1000 g cm-2. Is this just a scenario that could explain the measured nuclide concentration assuming 100% production during burial?

Page 17, line 5: Please clarify where the burial age "near 0.5 Ma" comes from.

Technical corrections

Page 1, line 14 and Page 17, line 27: add OLD after 260 ka.

Page 2, line 12: remove the last "the"

Page 5, line 5: San Gorgonio Pass region and San Bernardina Mountains are not visible on Figs 2 and 3. For readers who are not familiar with these locations, more specifications about the geographic relationship between them and the study site will be helpful.

Page 5, lines 29-30: I guess "late Pleistocene alluvial fans" refers to the yellow signature in Fig. 2, which is called "late Quaternary" in the legend. To facilitate reading, please be consistent between text and legend.

Lines 30-31: avoid using two times "estimated"

Page 10, line 15: estimating

Line 24: Note that this parameter is the production rate of epithermal neutrons (from

fast neutrons) in the atmosphere

Fig. 6, right panel: For better readability, I suggest writing some of the names, e.g. those of the surface samples, on the left side of the curves.

Fig. 7, caption: sample MCP-11a does not exist, it should be b. For clarity, put the last sentence of this caption before "Note that the steady erosion. . .".

Page 13, lines 7-8: "a possible explanation for this mismatch is that" could be removed

Page 14, lines 7-11: this sentence is too long and therefore hard to read. Does "highlights" refer to "agreement between the calculated and observed 36Cl/10Be ratio. . ."? If yes, it doesn't seem logical.

Table 1: the caption only describes parts of what is shown in the Table

---

## Referee Comment (RC2) · Jennifer Lamp (Referee) · 4 Jun 2019

General Comments

This paper by Balco et al. is a thorough investigation into the use of the 36Cl/10Be cosmogenic nuclide pair (produced in the minerals K-feldspar and quartz respectively in the same granitoid clast) in burial dating. They argue that this pair is more accurate than the commonly-applied 26Al/10Be pair for sediments/clasts in the range of 200-500 kyr due to the shorter half-life of 36Cl compared to 26Al. In the study, the authors use the technique to uncover information about the age of sediments displaced by a portion of the San Andreas fault in southern California. While their conclusions are

complicated by low nuclide concentrations and difficulties in estimating the 36Cl/10Be production ratio for the buried clasts, I find the description of the technique and the authors' comprehensive look at multiple aspects of dating clasts with this novel method to be particularly illuminating and extremely useful to the wider cosmogenic exposure dating community. They investigate quantitatively possible explanations for the scatter in their dataset, in addition to general limits and applicability of the 36Cl/10Be burial dating pair. I additionally appreciate the authors including the detailed AMS data calculation spreadsheets in the supplement, and providing their MATLAB scripts online for all to reference.

My only general critique is that I'd like to see a more detailed discussion of the uncertainties associated with using this technique, and the impacts on the final 36Cl/10Be ratio. Should "external" errors be used (i.e., those including uncertainties in the productions rates in addition to measurement uncertainties) because you're comparing two different isotopes in two different minerals with varying production pathways? While I'm not an expert on 36Cl cosmogenic dating, I would expect that the multiple production paths for 36Cl and the 36Cl production rate dependence on the chemical makeup of the K-feldspars and bulk rock (plus the uncertainty in water content, etc.) could make the error on the 36Cl concentration (and hence the final ratio) quite large depending on what uncertainties are propagated through the calculations. It's possible that this information could be gleaned from the MATLAB scripts, but it would be nice to see a few sentences of discussion in the manuscript about this.

Overall, this is an excellent paper and I highly support it being published with only minor edits.

Specific Comments • I wonder if the title could be reframed to focus more on the technique than the specific Mission Creek application, as I think the study most convincingly explores the background and limitations of the 36Cl/10Be pair as a general burial dating technique. Something like: 36Cl/10Be burial dating of granitoid clasts: a case study in the San Andreas Fault system (etc.). Or, something that would highlight

the technique/method over the application in this case.

• Page 2, Eq. 3 and Lines 2-5: Eq. 3 and the variables therein are difficult to interpret at first glance due to the lack of subscripting.

• Page 2 Lines 10-15: See general comment above; does the better precision for the $^{36}Cl/^{10}Be$ pair hold out if production rate/chemical composition uncertainties are taken into account?

• Page 6, Lines 8-10: Do you have pictures of these samples? Perhaps include them in the supplement if not in the main text?

• Page 7, Lines 11-12: Supplementary tables 1 and 2, or supplementary spreadsheets 1 and 2 (SF1, SF2)? I don't see specific supplementary table names in the files.

• Page 8, Lines 2-3: How does the amount/uncertainty of Cl in the HF affect the resulting burial age uncertainty?

• Page 8, Line 5: Table S3 = spreadsheet SF3?

• Page 10, Lines 17-24: What are the uncertainties on these production rates?

• Page 11, Lines 6-7: It would be interesting to provide a plot of either the $^{36}Cl/^{10}Be$ production ratio or just the $^{36}Cl$ production rate vs. K-concentration for each sample as part of Fig. 5 or 6. (The reader could glean this from info in the Tables, but it would be nice visual).

• Page 14: Line 31: "...do not show evidence of significant burial" is a little confusing because you also assert that the $^{36}Cl/^{10}Be$ ratios are due to post-burial nuclide production. Perhaps rephrase slightly?

• Page 16, Lines 9-14: Are there any visual differences (weathering features, grain size, etc.) between sample MC-P7-8 and the others?

  Page 16, Lines 14-15: Are these surfaces the same lithology as MC-P7-8?

  Page 17: Lines 10-11: Perhaps rephrase this; e.g. "For the minority of samples in this study that have relatively high nuclide concentrations, and possible arise from relatively low-erosion-rate environments…"

  Figure 2: Including a map here that is in between the scale of the inset regional map and the sample map would be helpful; it's a bit difficult to understand the position of the study site.

  Figure 6: Can you add sample labels to Panel A? The bold lines in this panel are difficult to discern; I assume they just all overlap?

  Figure 7: Caption should read MCP-11b instead of MCP-11a.

  Table 1: Are the sample thicknesses and densities listed somewhere? Also, there is a superscript missing for "g-1" under "cosmogenic 36Cl".

  Table 3: extra space before the period in the table title.
* * *

---

## Author Comment (AC1) · 20 Jun 2019

Both reviews of this paper were very positive and recommended publication with minor revisions, but highlighted a number of sections of the text that were unclear or lacked needed details. We appreciate the careful attention to the paper by both reviewers, and here we respond to the comments in the first review by Irene Schimmelpfennig.

Note that this review included a number of comments that referred only to typographical errors or minor technical corrections. We have corrected these in the revised text and do not respond to them specifically here. Review comments are highlighted below in italics and our responses follow.

[Figure]

Specific responses:

*Line 22 – page 2, line 1: maybe clarify that "sediment is exposed to the surface cosmic-ray flux during erosion" refers to processes in the source area of the material and also includes bedrock.*

This is correct. We clarified it.

*Page 2, 7-9: It would be appropriate to be a bit more specific, i.e. to give examples of the methods that you compare the cosmogenic nuclide burial dating with and add references. Do you mean to imply here that burial dating is only useful in arid regions?*

We did not mean to imply this. We have clarified this section of the text.

*Page 3, line 17-18: Bierman et al., GCA (1995) attempted to use cosmogenic 36Cl produced from Cl in fluid inclusions in quartz to quantify erosion rates and exposure ages. I think it would be appropriate to cite this study.*

We agree, and we appreciate the reviewer's reminding us of this study, which we had totally forgotten about.

*Page 5, line 28: "dated to 0.5 – 1.1 Ma" using which method?*

This age assignment relies on a variety of lines of evidence, including magnetostratigraphy, tephrostratigraphy, and stratigraphic correlations, that are described in detail in the paper by Kirby and others that is cited in this sentence. As the present paper is not about the Ocotillo Fm. and the purpose of this sentence is simply to note that previous mapping has assumed that the alluvial fan sediments in our field area are Plio-Pleistocene because they are generally similar to known Plio-Pleistocene sediments in the region, we have not added any additional discussion here.

*Page 5, Line 33 – Page 6, lines 2: I guess the samples from the three stratigraphic levels were taken along a horizontal transect and not on a vertical transect due to issues of accessibility?*

The specific sample sites for each stratigraphic level were mainly chosen so that we could sample from sites where there was evidence for rapid erosion, thus minimizing the possibility of significant cosmic-ray exposure during exhumation of the sites. We were best able to achieve this by sampling from a series of rapidly eroding stream banks and valley walls located along a channel incised into the alluvial fan system.

*Page 7 line 14: "a double-isotope-dilution method": I guess you refer to the routinely used isotope dilution technique that is described e.g. in Ivy-Ochs et al, NIMB (2004) and Desilets et al., Chemical Geology (2006)? Otherwise please clarify.*

This is correct in general terms, although we are not certain that our procedure is identical to theirs in all respects. Thus, we did not refer to a single previously published description, but instead described our laboratory procedure in detail, including the characteristics of the isotopically enriched spike that we used. This description is in lines 15+ of page 7 of the submitted text. In addition, the supplemental material contains additional characterization of the spike as well as all the isotope dilution calculations.

*Page 8, line 7: in which SGS lab the aliquots were analyzed?*

SGS Mineral Services, Lakefield, Ontario, Canada.

*Page 9, line 6: Please note that thermal neutrons are directly produced during spontaneous fission of U, but indirectly during decay of U and Th through alpha,n-reactions (see Gosse and Phillips, 2001; therein called "nucleogenic" instead of radiogenic 36Cl, which would be probably more correct)*

Correct. "Nucleogenic" is more accurate. Corrected in revised text.

*Page 10, lines 1-2 and 16: Please quantify the "small effect" of these different muon-related scaling differences. E.g. according to the models you use for the calculations, what are the muogenic 10Be and 36Cl contributions in the range of altitudes of the considered catchments?*

In the submitted paper, we did quantify the uncertainty in burial ages associated with not knowing the source elevation of a sample, in line 30 of page 16 of the original text. In the revised text we have added a reference to the later discussion.

*Page 11, line 8: for completeness it could be added that 36Cl production from K spallation and 10Be production in quartz are assumed to scale identically.*

This was stated in the submitted paper in line 17-18 of page 10.

*Lines 9-11: Here again, please quantify the "significant fraction" of 36Cl produced from muons in the considered depth range. This sentence seems in contradiction with lines 1-4 of the same page, where you imply that the muon-related 36Cl production and associated inaccuracies in the considered subsurface are low enough to be insignificant. Please clarify.*

We agree that this section of the text would benefit from some clarification, and we have done this in the revised text. Basically, there are two issues here that we have mixed up.

If we can assume that a sample that we are trying to burial-date was initially exposed at the surface, then uncertainties in muon production rates are insignificant, because muon production makes up such a small fraction of surface production.

On the other hand, for the fluvial clasts with measured ratios that are higher than expected for surface exposure, one possible explananation is that the clasts were never exposed at the surface, but were exposed in the subsurface and delivered directly to the channel by landsliding. The reason the production ratio is higher in the subsurface is that muon production makes up a significant fraction of total production in the subsurface. However, the fact that we have no idea what depth the samples came from creates a much greater uncertainty in what we expect the source production ratio to be than any uncertainty in the muon production rates themselves. So in this situation we are very uncertain as to what the source production ratio is, but this is not because of

the uncertainty in muon production rates.

The important thing is that in neither situation do we really care about the uncertainty in muon production rates. It's either (i) insignificant, or (ii) significant, but much less important than other bigger problems.

*Lines 15-17: It should probably be added that the predicted production ratios depend on altitude and feldspar composition and are therefore different. Which sample-specific characteristics are included in the "calculated surface production rates" that you use to normalize the nuclide concentration: scaling, but also feldspar composition, right? I guess the calculations are those in equation 1?*

Again, we agree that this section of the text needs to be clarified. As this is a production-rate-normalized diagram, the simple exposure and steady erosion lines are computed for an idealized sample with surface production rates of both nuclides equal to one. The sample-specific production rate calculations are used to normalize the measured nuclide concentrations so that they can be plotted on the diagram. This operation is routine in the commonly used Al-26/Be-10 two-nuclide diagram, but this diagram is rarely used for Cl-36, so we agree that more explanation would be useful. We have added more explanation.

*Page 11, line 17-Page 12, line 2: What would happen if the samples were not steadily eroded? On page 16, lines 18-19, you seem to imply that if the sample didn't originate from a steadily eroding surface, the burial age could not be determined – probably due to the varying production rate ratio? Is this issue unique to the 36Cl-10Be pair?*

We did not mean to imply this. "Steady erosion" should read "surface erosion" and we have so corrected the text.

*Page 12, lines 9-11: this seems like a circular argument: you use the comparison of the observed to calculated ratio to check whether or not the sample is at steady state erosion (lines 3-5), and here you use the same comparison to conclude that the applied*

*reference production rates are accurate. Please clarify.*

Once again, we agree that this was not clear in the submitted text. The correct line of reasoning is: (i) the geological context of the sample implies that it has experienced an extended period of surface erosion; (ii) thus, the sample data must lie on the steady erosion line in a correctly constructed two-nuclide diagram; (iii) as we observe that the data do lie on the steady erosion line, the diagram must be correctly constructed, i.e. our production ratio calculation must be correct. We revised the text to better communicate this.

*Fig. 7, caption: Regarding the sentence that starts with "Note that the steady erosion. . .", please specify "high erosion rates". According to the diagram, the sim- ple exposure line lies below the steady erosion line from the left-most erosion rate (âĹ́ij15 m/Ma) on, so over the whole spectrum of considered erosion rates.*

We corrected this by increasing the x-axis range so that the crossover of the simple exposure and steady erosion lines is clearly evident.

*Page 16, lines 3-5: Please clarify how you determined 0.2 Ma burial at a depth of 1000 g cm-2. Is this just a scenario that could explain the measured nuclide concentration assuming 100% production during burial? .*

Yes, this is a representative possible scenario that we use to highlight the fact that even a very small amount of exposure during exhumation could account for a lot of the measured nuclide concentrations. Of course many other similar scenarios are possible. We have clarified this in the text.

*Page 17, line 5: Please clarify where the burial age "near 0.5 Ma" comes from.*

This is an approximate estimate that can easily be obtained from the two-nuclide dia- grams in Fig. 6. We have clarified this in the text.

---

## Author Comment (AC2) · 20 Jun 2019

Both reviews of this paper were very positive and recommended publication with minor revisions, but highlighted a number of sections of the text that were unclear or lacked needed details. We appreciate the careful attention to the paper by both reviewers, and here we respond to the comments in the second review by Jennifer Lamp.

This review included (i) one general remark about the paper overall; (ii) several comments on specific sections of the paper; and (iii) minor typographical errors or technical corrections. We respond to (i) and (ii) here. We have corrected the minor errors in the revised text and do not discuss them specifically here.

[Figure]

Review comments are highlighted below in italics and our responses follow.

*My only general critique is that I'd like to see a more detailed discussion of the uncertainties associated with using this technique, and the impacts on the final 36Cl/10Be ratio. Should "external" errors be used (i.e., those including uncertainties in the productions rates in addition to measurement uncertainties) because you're comparing two different isotopes in two different minerals with varying production pathways? While I'm not an expert on 36Cl cosmogenic dating, I would expect that the multiple production paths for 36Cl and the 36Cl production rate dependence on the chemical makeup of the K-feldspars and bulk rock (plus the uncertainty in water content, etc.) could make the error on the 36Cl concentration (and hence the final ratio) quite large depending on what uncertainties are propagated through the calculations. It's possible that this information could be gleaned from the MATLAB scripts, but it would be nice to see a few sentences of discussion in the manuscript about this.*

We certainly agree with the gist of this comment – that in certain situations, production rate uncertainties for some Cl-36 production pathways can be very large, which would result in terrible precision for burial dating with nuclide pairs including Cl-36. We can try to clarify some additional aspects of the comment here.

In most cosmogenic-nuclide literature, this paper included, "internal" uncertainties in a burial age refer to uncertainties stemming only from measurement uncertainties in nuclide concentrations, and "external" uncertainties are larger and take into account uncertainties in the independently determined parameters needed to compute the burial age, specifically the production ratio and the decay constants. This is true for burial dating with any nuclide pair, and is not specific to Cl-36.

Cl-36 production has some complications that are not present for Be-10 and Al-26, and these affect both of these uncertainties. First, the need to correct for supported nucleogenic Cl-36 adds uncertainty to measurements of the cosmogenic Cl-36 concentration, and therefore to both the internal and external uncertainties in a burial age.
However, as is very well highlighted in this study, this is only really important when cosmogenic Cl-36 concentrations are very low. Second, although estimates of spallogenic production rates for Cl-36 likely have similar precision as those for Be-10 and Al-26, estimates of thermal neutron production rates are extremely imprecise, so a sample with significant neutron capture production might have a very uncertain production ratio and therefore a very large external uncertainty. And then, finally, these two issues are linked in a complicated way, because high Cl concentrations in a target mineral lead to both high nucleogenic Cl-36 concentrations and high thermal neutron production; the former is only important when cosmogenic Cl-36 concentrations are low, but the latter is important always.

However, our main point in this paper is that both nucleogenic Cl-36 and thermal neutron capture production are serious problems – we discussed this at some length in pages 3-4 of the submitted paper – so we used feldspar separates with very low Cl concentrations and therefore very low neutron capture production, which minimize both problems. Having done this, we felt that it was off topic to provide a detailed review of exactly why they are serious problems (for example, the Alfimov and Ivy-Ochs paper gives an excellent review of the uncertainty in neutron capture production). Regardless, we appreciate the reviewer's calling our attention to this, and have added some material to this discussion in an attempt to clarify this issue.

*I wonder if the title could be reframed to focus more on the technique than the specific Mission Creek application, as I think the study most con- vincingly explores the background and limitations of the 36Cl/10Be pair as a general burial dating technique. Something like: 36Cl/10Be burial dating of granitoid clasts: a case study in the San Andreas Fault system (etc.). Or, something that would highlight the technique/method over the application in this case.*

Regrettably, we disagree. We think that the technical aspects of the burial-dating method and the slip rate of the San Andreas Fault system are both important. We tried several titles and we think this one is the clearest and most compact.

*Page 2 Lines 10-15: See general comment above; does the better precision for the 36Cl/10Be pair hold out if production rate/chemical composition uncertainties are taken into account?*

Yes, as long as Cl-36 production is nearly all by Ca and K spallation. That's the assumption in Figure 2. We clarified this.

*Page 6, Lines 8-10: Do you have pictures of these samples? Perhaps include them in the supplement if not in the main text?*

Unfortunately, we do not have suitable photos of all the samples. We agree that this would have improved the paper.

*Page 8, Lines 2-3: How does the amount/uncertainty of Cl in the HF affect the resulting burial age uncertainty?*

It's just lumped into the uncertainty in the overall measurement of the Cl concentration, because we measured it and corrected for it. We clarified this in the text. It is important to note that in a situation like this one where total Cl concentrations are very low, failing to correct for this would eventually lead to large errors in the nucleogenic Cl-36 estimates.

*Page 11, Lines 6-7: It would be interesting to provide a plot of either the 36Cl/10Be production ratio or just the 36Cl production rate vs. K-concentration for each sample as part of Fig. 5 or 6. (The reader could glean this from info in the Tables, but it would be nice visual).*

Because nearly all production in these samples is from K spallation, the relationship is basically just a straight line. The plot is attached to this response as Fig 1, but we didn't think it justified an additional figure in the paper.

*age 14: Line 31: ". . .do not show evidence of significant burial" is a little confusing because you also assert that the 36Cl/10Be ratios are due to post-burial nuclide production. Perhaps rephrase slightly?*

We agree this is unclear. We have tried to clarify it in the revised text.

*Page 16, Lines 9-14: Are there any visual differences (weathering features, grain size, etc.) between sample MC-P7-8 and the others?*

Unfortunately, no. If there were any identifiable differences, we might have had a better success rate. This remains a serious obstacle to using burial dating in this geologic environment.

*Page 16, Lines 14-15: Are these surfaces the same lithology as MC-P7-8?*

Unclear, because the Binnie erosion rate estimates are for whole catchments. This is intended only to show that the order of magnitude of the erosion rates is within that observed in previous studies generally.

*igure 2: Including a map here that is in between the scale of the inset regional map and the sample map would be helpful; it's a bit difficult to understand the position of the study site.*

This is true. We have improved the map.
* * *
**Fig. 1.**

[Figure]